



# Development of compact continuous measurement system for atmospheric carbonyl sulfide concentration

Kazuki Kamezaki[1], Sebastian O. Danielache[2], Shigeyuki Ishidoya[1], Takahisa Maeda[1], Shohei Murayama[1]

[1]National Institute of Advanced Industrial Science and Technology (AIST), Tsukuba 305-8569, Japan

[2]Faculty of Science & Technology, Sophia University, Chiyoda-ku, Tokyo 102-8554, Japan

*Correspondence to*: Kazuki Kamezaki (kamezaki-k@aist.go.jp)

**Abstract.** Carbonyl sulfide (COS), the most abundant sulfur-containing gas in the atmosphere, is a source of stratospheric sulfate aerosol (SSA) and is used as a tracer for gross primary production (GPP). However, tropospheric COS sources and sinks entail great uncertainty due to the limited number COS observation sites. Thus, field measurements of COS
concentrations worldwide are necessary to estimate the contribution of SSA and the global scale of GPP. Recently, MIRA Pico, a portable continuous COS concentration analyzer using mid-infrared absorption, has been released. MIRA Pico has a lower cost and is smaller than conventional laser COS analyzers. We modified and tested the MIRA Pico for atmospheric COS concentration measurements. The modified MIRA Pico exhibited $\pm$ 7.9 picomol (pmol) mol$^{-1}$ (1$\sigma$) for a 15-min average, and calibration gas consumption was as low as no more than 3 L d$^{-1}$. We also used the modified MIRA Pico for observations at
Tsukuba, Japan. The observed COS concentrations ranged from 425 to 604 pmol mol$^{-1}$, averaging a standard deviation (1$\sigma$) of (505 $\pm$ 33) pmol mol$^{-1}$. The observed values agree with previous observations and exhibit clear diurnal variations. Furthermore, we installed the modified MIRA Pico in a passenger car to observe the COS concentration distribution in Tsukuba City.

## 20   1 Introduction

Carbonyl sulfide (COS or OCS) is the most abundant sulfur-containing gas in the ambient atmosphere, with an average concentration of approximately 500 picomol (pmol) mol$^{-1}$ in the troposphere (Chin and Davis, 1995; Montzka et al., 2007). Because of its long lifetime (more than 2 yr), COS is converted to stratospheric sulfate aerosol (SSA) in the stratosphere (Crutzen, 1976), affecting the Earth's radiation balance and ozone depression. Furthermore, COS has been suggested as a
potential tracer of gross primary production (GPP) because of a similar uptake mechanism into plants through stomata as $CO_2$ but is not re-emitted by plants (Sandoval-Soto et al., 2005). Therefore, COS is recognized as a proxy for estimating GPP in ecosystems wherein carbon-climate feedback is the largest and most uncertain (Campbell et al., 2008). However, despite this importance, tropospheric OCS sources and sinks entail great uncertainty due to the limited number COS observation sites. Thus, field measurements of COS concentrations worldwide are necessary to estimate local contributions to SSA production
and the global scale of GPP.

Historical COS concentration measurements have been based on individual (flask) samples analyzed by gas chromatography–mass spectrometry (GC–MS) (Montzka et al., 2007). Although this method provides high precision, it is



characterized by low temporal resolution. Furthermore, stability within the flask is an issue for COS. Since around 2010, *in situ* COS concentration measurements using laser spectroscopy have been reported. The Aerodyne COS analyzer (Aerodyne

Research Inc.) provides high-frequency (over one Hz) COS and $CO_2$ measurements (Stimler et al., 2010a, b; Commane et al., 2013; Belviso et al., 2013; Berkelhammer et al., 2014; Kooijimans et al., 2016; Belviso et al., 2020; Zanchetta et al., 2023). Later, the Los Gatos Research (LGR) COS analyzer (LGR COS analyzer) using cavity ring-down spectroscopy was also used to measure COS concentrations, enabling COS and $CO_2$ measurements at high frequencies (above one Hz) (Berkelhammer et al., 2014; Belviso et al., 2016; Rastogi et al., 2018). However, these devices are large, expensive, and costly to maintain.

Therefore, it is desirable to develop a compact and inexpensive continuous measurement device capable of COS measurements with good precision (below 15 pmol mol$^{-1}$) at a low calibration frequency. If the device is compact and low-cost, it can be used for multipoint observations and installed on mobile objects (such as cars or planes) easily. Observation of COS concentration using mobile objects is useful for clarifying the spatial distribution of COS concentration (Berry et al., 2013; Zanchetta et al., 2023). These efforts are expected to improve our understanding of the spatiotemporal distribution of COS

concentrations, which will be essential for future inverse analysis of source distributions.

Recently, MIRA Pico (Aeris Technologies, CA, USA), a portable continuous COS concentration analyzer using mid-infrared absorption, has been released. The price of this analyzer is less than half that of a conventional laser COS analyzer, and the device is small. However, MIRA Pico has never been applied to actual long-term observations of COS concentrations. Therefore, we aimed to improve this analyzer so that it can be used for actual atmospheric observations. Furthermore, we

installed the modified MIRA Pico in a passenger car and observed the COS concentration in Tsukuba City in Japan.

## 2 Materials and Methods

### 2.1 Measurement system

COS concentrations were measured using a MIRA Pico system. The MIRA Pico is portable, field-deployable, and capable of

simultaneous measurements of COS, carbon dioxide ($CO_2$), and water concentrations. Briefly, the MIRA Pico is ultracompact (30 cm × 20 cm × 10 cm) and lightweight (3 kg) and uses a small cell (60 mL). This analyzer is equipped with two inlets, allowing for improved accuracy by exchanging the reference and sample gases at regular intervals.

MIRA Pico determines the $CO_2$ and COS spectra by referring to the position of the water spectral peak. Therefore, a certain amount of water (>5,000 μmol mol$^{-1}$) in air is essential for measuring COS concentrations. The MIRA Pico series is

characterized by compactness and is also used for the measurement of other compound gases such as methane (Commane et al., 2023). However, no COS concentration measurements using the MIRA Pico have been previously reported. On its default settings, this system exhibits a significant signal drift, with a standard deviation of 10-min accuracy (1σ) of ± 50 pmol mol$^{-1}$, and accuracy is also sensitive to ambient temperature. To address these issues, the system was modified, as shown in Figure 1. The solenoid valve, filter, pressure valve, and mini pump of the MIRA Pico analyzer remained unchanged. The flow rate

can be controlled by changing the voltage of the pump, which was set to approximately 210 mL min$^{-1}$. The pressure inside the optical cell was controlled at 140 mbar. The optical cell was obtained from an original MIRA Pico to control the temperature

of a commercial refrigerator set at 15°C (Figure 1). Inside the refrigerator, the optical cell was surrounded by thermal insulation, and the temperature was controlled using a Peltier cooler set at 29°C placed at the bottom (Figure 1). A cushioning material was placed between the Peltier cooler and the optical cell to minimize temperature fluctuations. The sample and reference

gases were switched every 30 s using a solenoid valve. Air flows through a filter, a pressure control valve, an optical cell, and a pump in MIRA Pico.

## 2.2 Inlet system

Air was pumped by a Teflon-coated diaphragm pump (D-79112, KNF, Freiburg, Germany) to Tee union through a 7-μm filter

(SS-4TF-7, Swagelok, State of Ohio, USA), passed through a 10-position valve, an electric cooler unit (ECU) set at 2°C, outside the Nafion dryer, and injected into port 1. The air separated at Tee passed through the ECU, activated charcoal, and Nafion dryer and was then injected into port 0 as reference gas. To investigate the effect of the ECU and pump on COS concentration, we installed a pump and ECU in front of port 1, opened port 0 to the atmosphere, and measured the COS concentration in the room air. Next, we performed a similar experiment with ports 0 and 1 reversed. When comparing the COS

concentration injected from port 1 and port 0 in each experiment, no change in COS concentration was confirmed with or without the pump and ECU. Also, it was also confirmed that the Nafion dryer had no effect on COS concentration, similar to the ECU and pump. Activated charcoal can remove part of COS. Conversely, water is not absorbed by activated charcoal, and it does not clog even after being used for more than three months. Because water tends to remain in the cell and the COS concentration appears to change because of the difference in water content between the reference and sample gases, the Nafion

dryer is installed in parallel to reduce the difference in water content between the sample and reference gas ports.

## 2.3 Standards

Ambient air gases in an aculife IV treated aluminum cylinder from NOAA/ESRL were used as COS standard gases. The COS concentrations of standard gases were $360 \pm 0.5$ (Standard A), $452.6 \pm 0.5$ (Standard B), and $565 \pm 0.5$ pmol mol$^{-1}$ (Standard

C), as determined using GC–MS by NOAA/ESRL (Montzka et al., 2007). Standard B was filled in 2021. Standards A and C were filled in 2022. Standards A, B, and C were connected to the multiposition valve as calibration gases. Because Standards A, B, and C were dry gases that could not be measured as they were, they were humidified by passing them through the ECU.

## 2.4 Data analysis

To ensure data stability, data collected 10 s after switching were disregarded. The measured value was obtained by subtracting the average value of the preceding and subsequent reference gases injected into port 0 from the average value of the gas injected into port 1.

## 3 Results and Discussion

### 3.1 Reproducibility of COS measurements


Allan deviation plots are an effective way to estimate the extent to which the effect of white noise is reduced and the effect of drift is increased by extending the measurement integration time (Allan, 1987). To evaluate the stability of the COS measurements, Standard C was injected to evaluate the Allan deviation without switching to the reference gas, and the measurement frequency was one Hz. Although many studies have reported that the COS concentration in the cylinder changes

slowly, it is unlikely that the gas in compressed air will fluctuate within a few hours. The Allan deviation plots of the original and modified MIRA Pico systems are shown in Figure 2. When the original system was used, a significant Allan deviation was obtained, ranging from 26.1 pmol mol$^{-1}$ at an integration time of 5 s to 212 pmol mol$^{-1}$ at 160 s. After modifications, the system exhibited an increase in the Allan deviation ranging from 16.6 pmol mol$^{-1}$ at 40 s to 38.7 pmol mol$^{-1}$ at 180 s, indicating that the modified system is more stable, as shown by the Allan deviation drift reduction. During this measurement, reference

gas was injected every 30 s to measure COS concentrations with high accuracy (low Allan deviation).

To check the repeatability of COS concentration measurements when injecting the reference gas every 30 s, we used Standard C as the sample gas and room air as the reference gas (Figure 3). Standard C passed through the ECU and Nafion dryer and was injected into port 1. Room air passed through the ECU, activated charcoal, and Nafion dryer and was injected into port 0 (see section 2.2). Direct measurements of COS concentrations using the modified MIRA Pico showed some level

of fluctuation (panel a). The Standard C measurement corrected with the reference gas injected every 30 s was stable with a standard deviation (1σ) of ± 22 pmol mol$^{-1}$ (panel b).

### 3.2 Accuracy of COS concentration measurement

A calibration curve was generated over a wider concentration range than previous countenious COS measurement methods

from the output for approximately 1.5 months using Standards A, B, and C. These three standard gases were injected every 10 min through a 10-port multiposition Valco valve. Then, the average value for the last 5 min was adopted as the value for each standard gas. The outputs of standard gases versus time and calibration curves are shown in Figures 4 and 5. The outputs of Standards A, B, and C were constant from April to May 2023 (Figure 4), showing no long-term accuracy drift during COS measurements, using our modified MIRA Pico system. The slope and intercept of the calibration curve were 0.90 ± 0.02 and

−17.2 ± 8.4, respectively (Figure 5). The 1σ of the residual between the linearity line constructed using the least-squares method and the measured value was ± 13.7 pmol mol$^{-1}$, and the correction efficiency (*r*-value) is 0.98 (Figure 5). Therefore, the calibration curve was linear over a wide range of 360–565 pmol mol$^{-1}$. For COS concentrations outside the range of the calibration curve, COS concentrations were measured by extrapolation. No more than 3 L of standard gas every 10 min was required, and a calibration frequency of one a day or less was sufficient.


### 3.3 Water vapor interference correction

We evaluated the effect of water content on COS concentrations. The water contents when Standards A, B, and C are introduced during the measurement period are shown in Figure 6a. The correlation between water content and Standards A, B, and C is shown in Figure 6b. Although the water content increased from 5,000 to 16,000 µmol mol$^{-1}$ from April to May 2023,





the outputs of Standards A, B, and C did not change with the change in water content. The water content of 5,000–16,000 µmol mol⁻¹ can correspond to a dew point of −2°C–20°C. The actual water content at the Tsukuba site was 4,000–46,000 µmol mol⁻¹ from April to May 2023. This shows that MIRA Pico programmatically corrects for water content. Therefore, we can conclude that atmospheric water content does not affect COS concentration measurements within a 5,000–16,000 µmol mol⁻¹ range.

According to the test results, the accuracy test includes a drift effect, repeatability, and water vapor interference. Therefore, we adopted the overall uncertainty (1σ) of this developed method of ± 13.7 pmol mol⁻¹ for 5-min measurements. The observed COS concentrations were calculated by integrating for 15 min, and the uncertainty (1σ) was assumed to be ± 7.9 pmol mol⁻¹ (13.7 divided by root 15/5).

**3.4 Comparison with other methods for continuous COS concentration measurements**

Continuous COS concentration measurements using Aerodyne and LGR COS analyzers have previously been reported (Stimler et al., 2010a, b; Commane et al., 2013; Belviso et al., 2013; Kooijimans et al., 2016; Berkelhammer et al., 2014; Belviso et al., 2016; Rastogi et al., 2018; Belviso et al., 2020). Regarding the uncertainty (1σ) of the measurements, Koiijimans et al. (2016) demonstrated that the overall uncertainty was ± 7.3 pmol mol⁻¹ for 1 min using an Aerodyne COS analyzer.

Berkelhammer et al. (2014) demonstrated that the repeatability uncertainty using a standard gas was approximately ± 7.5 pmol mol⁻¹ for 1 min using an LGR COS analyzer. Conversely, the overall uncertainty of our developed method exhibited ± 13.7 pmol mol⁻¹ for 5 min, and repeatability was ± 22 pmol mol⁻¹ for 1 min using a standard gas (see section 3.2). Therefore, the uncertainty of COS concentrations in this study is larger than that of previously reported COS analyzers. For the modified MIRA Pico, the temperature in the optical cell could only be controlled to approximately ± 0.5°C. If the temperature in the

optical cell could be controlled more precisely (as in Kooijmans et al., 2016), this system could produce more accurate measurements.

Calibration is important for accurate COS concentration measurements. Belviso et al. (2020) showed a significant drift of 180 pmol mol⁻¹ d⁻¹ using an Aerodyne COS analyzer without the injection of pure nitrogen gas. For the amount of calibration gas used, Kooijmans et al. (2016) showed that reference gas measurements using pure nitrogen gas every 30 min

are sufficient to reduce drift to within 5.3 pmol mol⁻¹ with temperature changes up to at least 0.06°C per 30 min, and the amount of calibration gas (zero gas) used was approximately 76 L d⁻¹, with every 30 min for 10 min at 160 mL min⁻¹ using an Aerodyne COS analyzer. Similarly, Rastogi et al. (2018) injected reference gas for 5 min at 1 L min⁻¹ daily, i.e., 5 L d⁻¹, using an LGR COS analyzer. Therefore, the modified MIRA Pico uses the least amount of calibration gas per cylinder per d (3 L per d).

In terms of electric consumption, the original Aerodyne COS analyzer uses 250 W (without pump), and the original LGR COS analyzer uses 400 W. Conversely, MIRA Pico uses only 15 W. In terms of weight, the original Aerodyne COS



analyzer is 35 kg (measurement part only), and the original LGR COS analyzer is 68 kg. Meanwhile, the modified MIRA Pico is only 10 kg (including all units).

Although the measurement uncertainty of the modified MIRA Pico is not as low as that of reported COS analyzers in the literature, it has the advantages of compactness, low drift, low power consumption, low weight, and low consumption of calibration gas. Therefore, the developed system is useful for observations in places with electricity and space limitations and for observations in cars wherein calibration gas preparation is difficult because it has different characteristics from previous COS analyzers.

## 4 Field Observations

### 4.1 COS concentration at Tsukuba site

Continuous observations of COS and $H_2O$ concentrations were performed at the National Institute of Advanced Industrial Science and Technology in Tsukuba, Japan (36.05°N, 140.12°E, 12 m above ground level) from April 11 to 21, 2023 (Tsukuba site in Figure 7). Tsukuba is a suburban area with low-lying land with no mountains or other obstacles to the east and south to

reach the Pacific Ocean at approximately 50 km, while the north and west is mountainous inland. Industrial areas, such as power plants, are located southwest of Tsukuba. During the observation period in Tsukuba City, the sunrise time was at 5:11 (Japan Standard Time (JST)), and the sunset time was at 18:10 (JST).

Figure 8 shows COS concentrations from April 11 to 21, 2023, at the Tsukuba site. The observed COS concentrations ranged from 425 to 604 pmol mol$^{-1}$, averaging with a standard deviation ($1\sigma$) of 505 ± 33 pmol mol$^{-1}$. Measured COS

concentrations are consistent with previous observations using isotope ratio mass spectrometry at Yokohama, Japan (Kamezaki et al., 2019 and Hattori et al., 2020) and with COS concentrations observed at similar latitudes in the USA (e.g., 400–550 pmol mol$^{-1}$; Montzka et al., 2007). Furthermore, the COS concentrations exhibited diurnal variations. They decreased at night, and after sunrise, they increased until approximately 16:00. However, after 16:00, they decreased. This diurnal COS concentration trend is consistent with the diurnal variation in COS concentrations observed at forest sites in the USA in August

(Berkelhammer et al., 2014), in France in March (Belviso et al., 2020) and in June (Belviso et al., 2016), over the sea in September–October (Berkelhammer et al., 2016), and in the USA in August–September (Rastogi et al., 2018). Therefore, the modified MIRA Pico can sufficiently measure daily variations in ambient COS concentrations. The decrease of COS at night can be thought to be mainly caused by soil bacteria, as reported by Kato et al. (2008) and Kamezaki et al. (2016). The increase in COS concentrations after sunrise is suggested to be due to photochemical generation and/or vertical mixing (Berkelhammer

et al., 2014, 2016; Campbell et al., 2017). However, it is difficult to identify the cause and quantitatively evaluate the factors of diurnal variations from this COS concentration data alone. To clarify COS diurnal variation factors in more detail, further data accumulation of COS concentrations, comparison with other gas concentrations, meteorological data such as solar radiation and temperature, and isotopic compositions, and reproduction by numerical simulations are required.

### 4.2 COS concentration map in Tsukuba City.





To clarify the distribution of COS concentration in Tsukuba City, we installed the developed system on a passenger car and observed the COS concentration. The observation campaign was performed on August 6, 2023. COS concentrations were calibrated before and after the campaign. A modified MIRA Pico was placed inside the passenger car, and electricity was supplied by 1,000 W batteries for 5 h. The system obtained air through a sampling line, with its inlet placed on the top of the
vehicle. During the observation, GPS live data synchronized with the time log of the modified MIRA Pico were collected.

Figure 9a shows COS concentrations over time, and Figure 10 shows COS concentrations plotted on a map. The COS concentration ranged from 381 to 460 pmol mol$^{-1}$ in this campaign. Almost the same COS concentration was observed when passing through the same location on the outbound and return trips (Figure 10). Low COS concentrations were observed on the Mount Tsukuba (north end) at 12:52 (JST) and near the swamp (south end) at 14:13 (JST). Especially, a decrease in COS
concentration was observed from urban areas to Mount Tsukuba (Figure 10). The highest concentration was observed in the swamp at 11:31 (JST). Weather data from the nearest observatory were used (Tateno in Figure 10). From 12:00 to 13:00 (JST), the average wind speed was 1.3 m s$^{-1}$ (Figure 9b) and the maximum instantaneous wind speed was 3.8 m s$^{-1}$ (Supporting info). From 11:20 to 11:40 (JST), the average wind speed was 4.8 m s$^{-1}$ (Figure 9b), the maximum instantaneous wind speed was 10.6 m s$^{-1}$, and the wind direction was south–southwest. From 14:00 to 14:20 (JST), the average wind speed was 2.9 m s$^{-1}$
(Figure 9b), the maximum instantaneous wind speed was 6.3 m s$^{-1}$, and the wind direction was south–southwest. The clear weather hours were 12:00–13:10 (JST), and the rainy weather hours were 11:00–11:30 and 13:20–14:00 (JST) (Figure 9). For the rest of the time, it was cloudy.

Although the COS concentrations increased from sunrise to 14:00 (JST) by diurnal variations, as described above (see section 3.6), the COS concentrations decreased from 12:00 to 13:00 (JST) during this campaign. The area around the
Mount Tsukuba has the most vegetation in Tsukuba City, and the air masses barely move because of the low wind speed. The spatial distribution suggest that COS concentrations may have decreased through photosynthesis. This is consistent with the reports that photosynthesis is the main sink of COS (Whelan et al., 2018). Near the swamp, both high and low COS concentrations were observed at 11:31 and 14:13 (JST), respectively. Although photoresponsive COS production has been reported in lakes (Du et al., 2017), due to the wet weather during this campaign, COS generation was unlikely. Also, it was
not thought that photosynthesis would have a large effect near swamps. Given the high wind speed, COS concentrations were thought to be affected by air mass transport. Although it was difficult to identify the cause of the COS concentration change using only the observation data of this study, we could observe the COS distribution in Tsukuba City. By expanding the coverage of observations in the future, the distribution of COS will be revealed in greater detail, resulting in improved COS measurement accuracy.

Both the source and sink of COS remain uncertain. To improve the accuracy of COS balance, it will be necessary to establish new observation points for COS concentration and to understand the spatial distribution of COS concentration. The modified MIRA Pico is significantly cheaper than conventional COS analyzers, is easy to carry, and can be installed in a car. In addition, the calibration frequency is low and the power consumption is low, making it suitable for installation in mobile obsejcts.  Therefore, it is possible to increase the number of observation points and take modified MIRA Pico to places where





COS concentrations are expected to be low or high. More ground observations, such as those in this study, will help clarify the sources and sinks of COS.

**5 Conclusions**

In this study, we modified the portable continuous COS concentration analyzer MIRA Pico. After the modification, the Allen deviation decreased, and the drifting improved. Measurement accuracy was calibrated using three standard gases with a wide concentration range (360-565 pmol mol$^{-1}$) purchased from NOAA. A 15-min averaging uncertainty for COS concentrations was ± 7.9 pmol mol$^{-1}$. The uncertainty was inferior to that of previous continuous ambient COS measurement systems. However, the price of modified MIRA Pico is less than half that of other analyzers, and its compactness makes it suitable for field measurements. Furthermore, the smaller cell provides a faster response and requires less gas, saving calibration gas consumption when using MIRA Pico.

We observed COS concentrations using the developed system for 10 days in April 2023 in Tsukuba, Japan. The observed COS concentrations ranged from 425 to 604 pmol mol$^{-1}$, averaging a standard deviation (1σ) of (505 ± 33) pmol mol$^{-1}$. The observed values are consistent with previous observations and exhibit clear diurnal variations. Furthermore, we measured the COS concentration distribution in Tsukuba City by installing the developed system in a passenger car. Through this campaign, it was expected that the COS concentration would decrease because of photosynthesis when moving from urban areas to Mount Tsukuba under weak wind conditions.

The developed system is significantly cheaper than conventional systems, is easy to carry, and can be installed in a car. Therefore, it is possible to increase the number of observation points and take modified MIRA Pico to places where the COS concentration is expected to be low or high. With additional ground observations, the sources and sinks of COS.

**Open Research**

**Data Availability**

The datasets generated and analyzed during the current study are available via this link: https://doi.org/10.5281/zenodo.8388504. Weather data from the Tateno site using this research is available on the Japan Meteorological Agency website: https://www.jma.go.jp/jma/indexe.html

Author contribution

KK, SOD, SI, TK and SM designed the analysis and interpreted the results. KK wrote the original draft. All authors reviewed and edited the draft.



## Conflict of Interest

The authors declare no competing financial interests.

## Acknowledgments

We would also like to thank Shohei Hattori at the Tokyo Institute of Technology, Japan (current address: International Center for Isotope Effects Research Nanjing University, China) for informing us about MIRA Pico. Weather station (Tateno) data were provided by Japan Meteorological Agency.

## Finantial support

This study was supported by Grants-in-Aid for Scientific Research 20J01445 (K.K), 22K18028 (K.K.), 20H01975 (S.O.D. and K.K.), 22H03739 (K.K. and S.I.), 22H00564 (K.K. and S.I.), and 22H05006 (S.I.) from the Ministry of Education, Culture, Sports, Science, and Technology (MEXT), Japan and by Joint Usage/Research Grant of the River Basin Research Center (2021–2022), Gifu University.

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





Figures

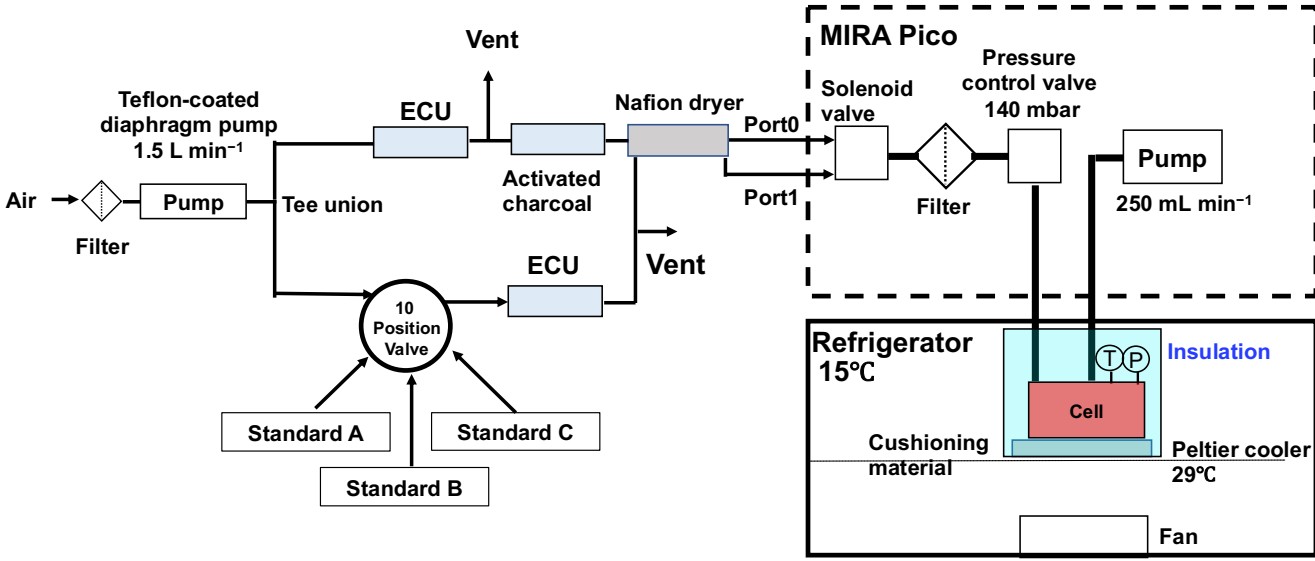

Figure 1 Schematic diagram of the modified MIRA Pico. System components: Pump, vacuum pump; cell, optical cell; ECU, electric cooler unit; T, temperature sensor; P, pressure sensor.



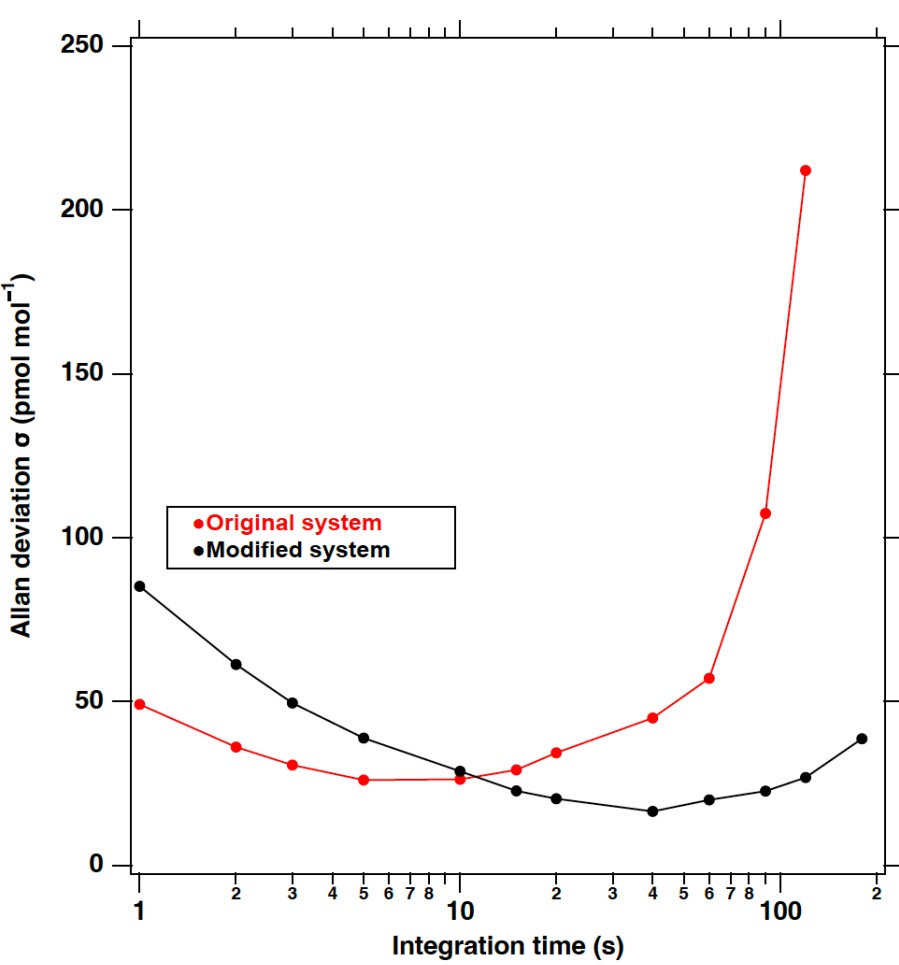

Figure 2 Comparison of Allan deviation plots original MIRA Pico (red) and after modification (black).





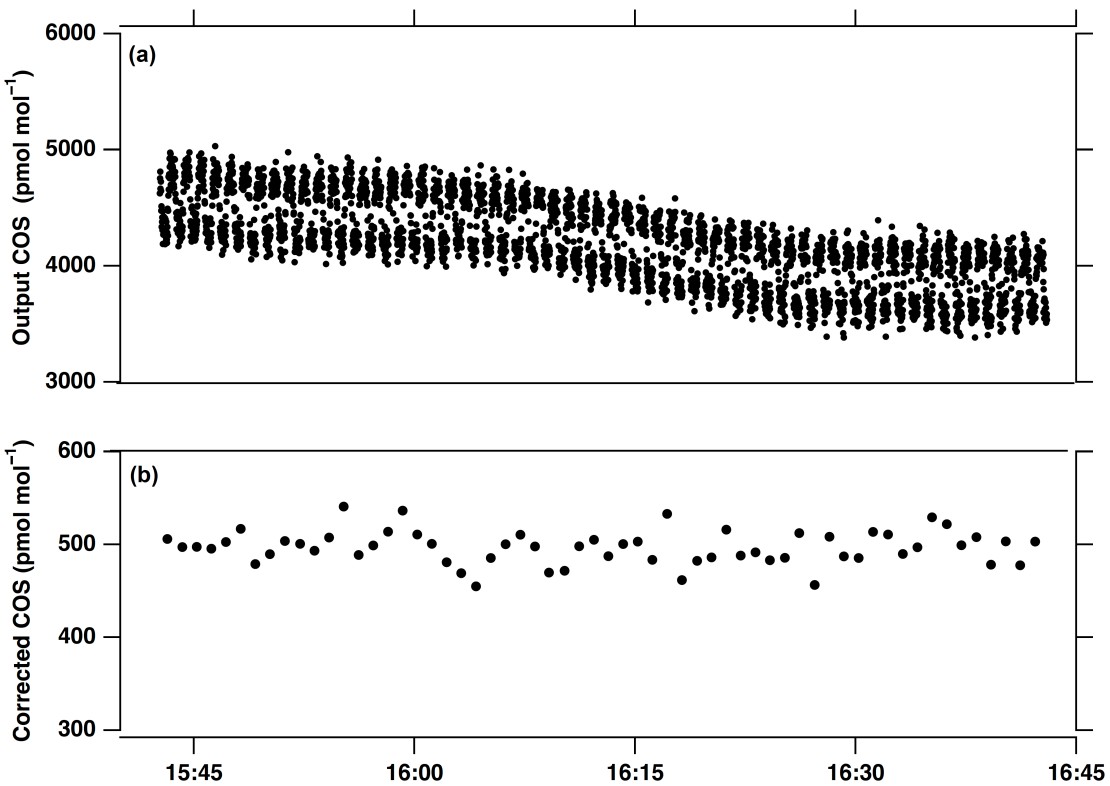

Figure 3 Evaluation of repeatability of COS concentration using standard C as a sample and indoor air as a reference gas. (a) direct output COS concentration from MIRA Pico. The plot was almost every second. (b) corrected COS concentration by reference gas. The plot was almost every minute.





Figure 4 Long-term trends in standard gases relative to reference gases.



Figure 5 Calibration curve of COS concentration. The standard deviation of the residual values gives the total uncertainty of the analytical method. Error bars are omitted, but there are errors of $\pm 9.9$ and $\pm 0.5$ pmol mol$^{-1}$ in the vertical and horizontal directions, respectively.





Figure 6 (a) Long-term trend of standard gases against water concentration and (b) correlation between water content and COS concentrations.



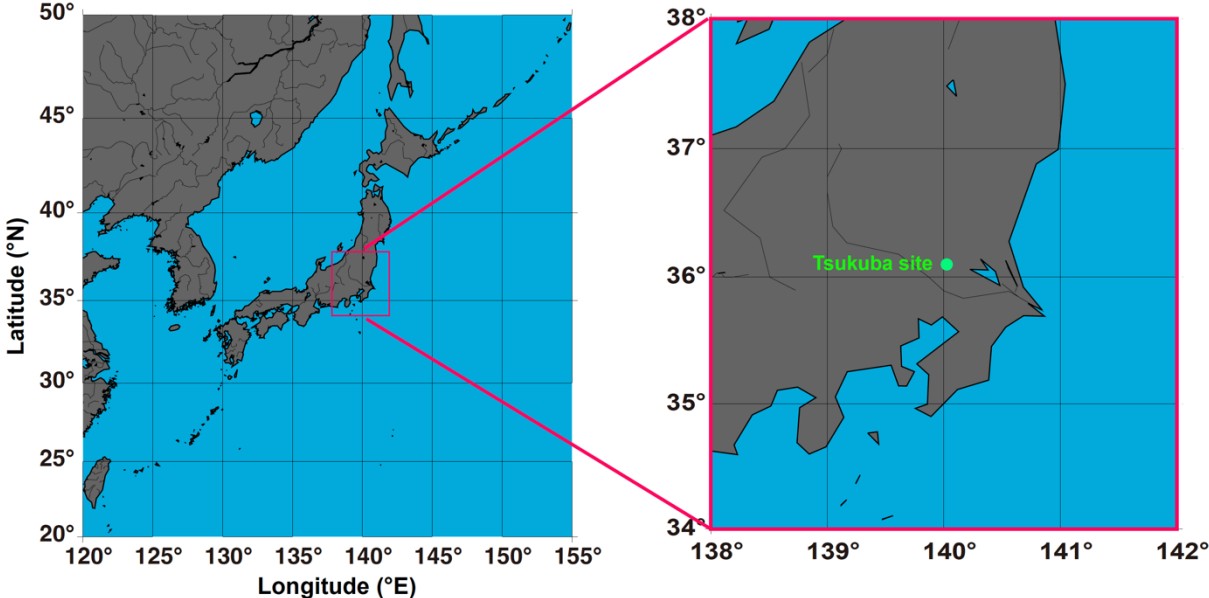

Figure 7 Aerial view of the sampling sites and the location of the Tsukuba site.

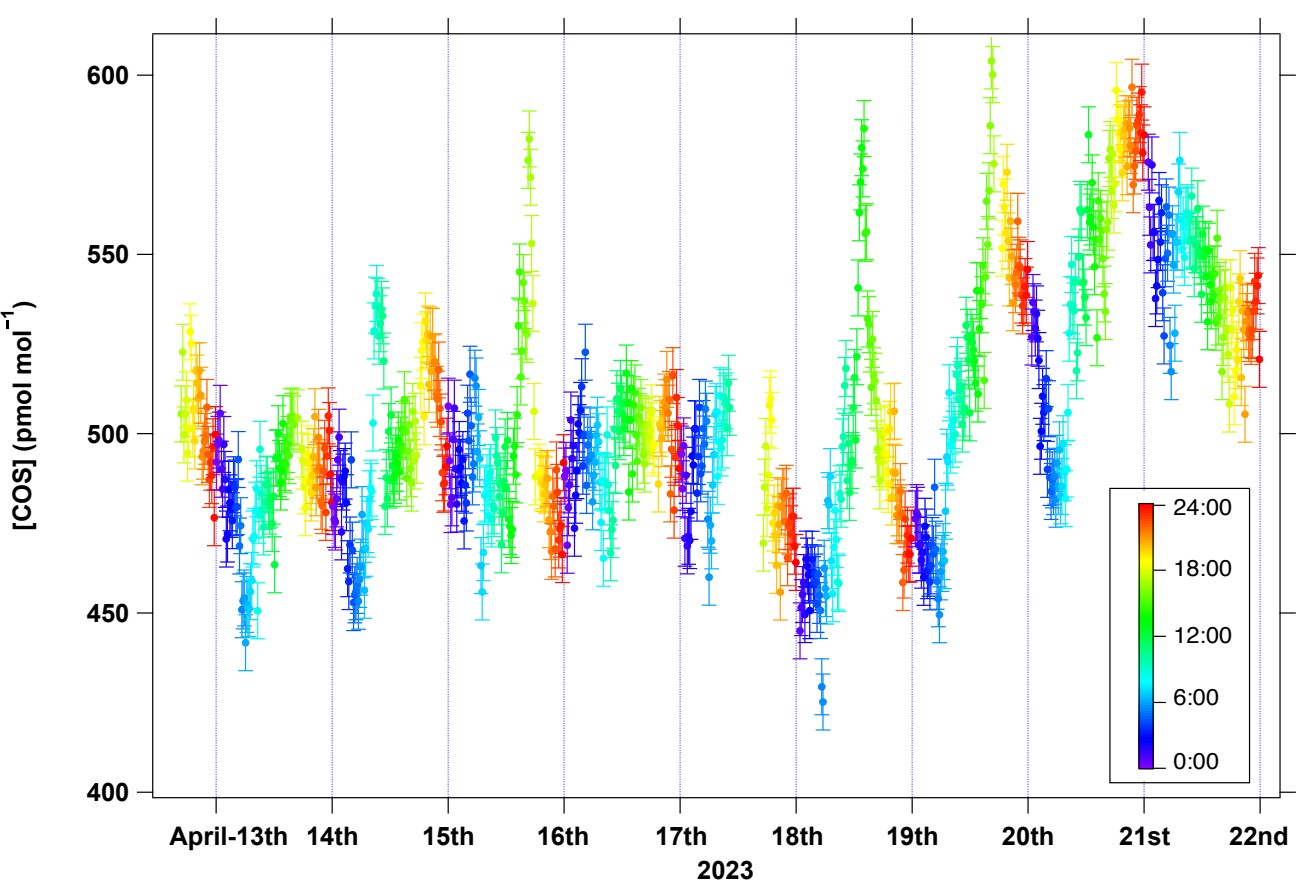

Figure 8 COS concentrations observed at the Tsukuba site. The color exhibits the time in a day.






Figure 9 (a) The COS concentration in the Tsukuba city observed using developed method putting on a passenger car. The concentrations are averaged for 15 minutes. The time is averaged time of 15 min sampling time. (b) The 10 min averaged wind speed observed at Tateno site. (c) The observed latitude at this campaign. (d) The observed Longitude at this campaign. The

blue line and red line show the time of rainy and sunny.






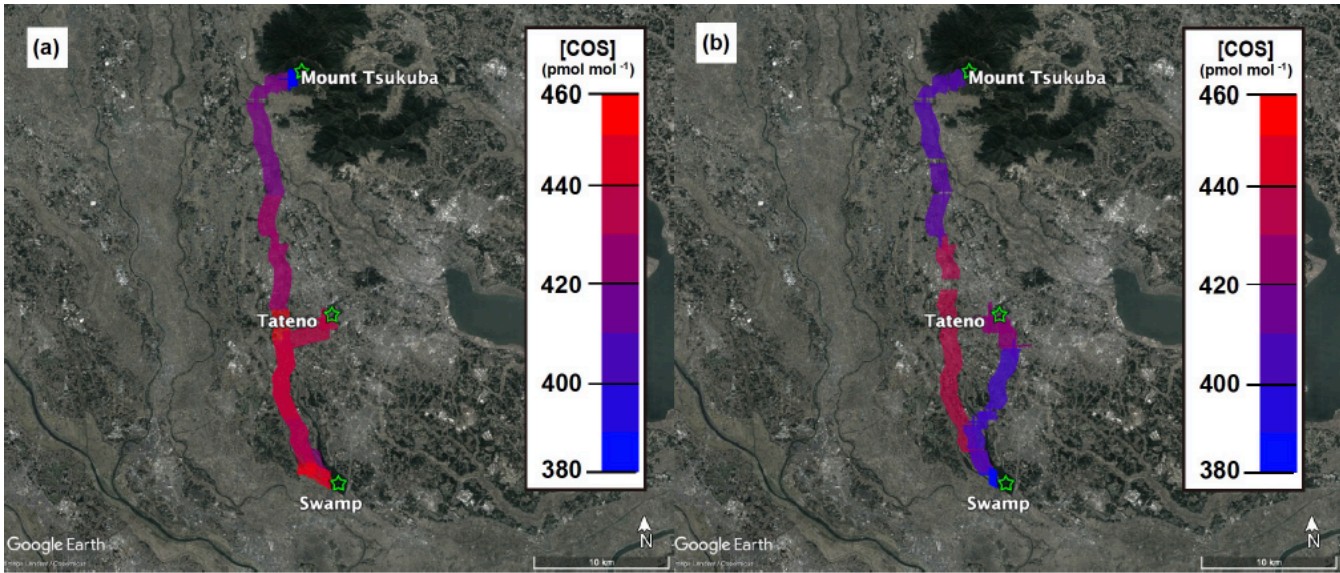

Figure 10 The average COS concentration over a 15 minutes colored on the map from © Google earth. Weather observatory station Tateno was also plotted. (a) Outbound route from Tsukuba site to Mount Tsukuba. (b) Return trip from Mount

Tsukuba to Tsukuba site.