# Peer review of "Development of compact continuous measurement system for atmospheric carbonyl sulfide concentration"

_Atmospheric Measurement Techniques, 2023_

## Author Comment (AC1)

Blue: Referee comment

Black: Our comment

Red: The sentences in our manuscript

General assessment:

Kamezaki et al present and test a new mid-cost analyzer (MIRA pico) for carbonyl sulfide (COS) concentration measurements. They also test a modified measurement setup in order to reduce the drift of the analyzer and report COS concentration measurements from Tsukuba, Japan.

COS measurements have gained more attention during recent years, especially because of the link between vegetation COS exchange and stomatal conductance and/or photosynthesis. However, the global COS budget is still not closed, partly due to the lack of a proper COS measurement network. As the mostly used gas analyzers (from Aerodyne and Los Gatos) are high cost, they are not that widely implemented. In the light of missing COS data, a mid-cost analyzer is very welcome to improve COS data availability. Given that the accuracy of the presented instrument is not very good compared to the Aerodyne and Los Gatos analyzers, I don't really see it as an option for atmospheric concentration measurements, that typically require very high precision. Instead, I would more likely see this type of portable analyzer convenient for e.g. chamber flux measurements. This study tests the long-term stability and reference gas consumption of the MIRA pico analyzer. However, the purpose of the use of reference gas is unclear, since the authors mention using room air as reference, that has an unknown amount of COS. The manuscript by Kamezaki et al still lacks many crucial information related to e.g. the measurement setup and the operation of the instrument they present, leaving the reader with many unknowns. The manuscript thus needs very substantial revisions and should either go through (very) major revisions, or rather rejected and resubmitted after modifications. I have listed the specific shortcomings below, followed by the detailed comments.

**Reply:** Thank you for your comment. We apologise for not providing sufficient information regarding the measurement method. We have carefully revised the manuscript based on your comments.

More information is definitely needed on the instrument itself: what is the size of the sample cell? How is COS concentration measured, at what wavelength and why does it need water vapor? Why is activated charcoal needed? Why a nafion dryer is installed if the measurement itself needs a certain amount of water vapor? How to know if there is enough water vapor for a reliable measurement? I also don't understand why there is an ECU to first humidify the sample air and after the ECU there is a nafion dryer to dry it…? What exactly is the refrigerator and how the sample cell can be moved there? What is the overall size of the modified system, is it still

portable? How can indoor air be used as reference gas as it has and unknown COS concentration? The authors need to be more specific on these details.

**Reply:** Thank you for your comment. The manufacturers were contacted for further information. We have added the following information to Section 2.1: COS concentrations were measured using a MIRA Pico analyser (Figure S1a). The MIRA Pico analyser is a portable, field-deployable instrument designed for the simultaneous measurement of COS, CO2, and water vapour concentrations with one Hz. The MIRA Pico analyser is ultracompact (30 cm × 20 cm × 10 cm), and lightweight (3 kg), uses a small cell (60 mL), and has an optical path length of 13 m. The upstream filter is a 0.01-micron fluorocarbon borosilicate glass microfiber element, and the tubing upstream from the optical cell is made of polyurethane tubing. The flow rate was adjusted using the pump voltage, which was set at approximately 210 mL min$^{-1}$. The pressure inside the optical cell was 140 mbar. The analyser incorporates a mid-infrared tuneable diode laser and a HgCdTe (MCT) photovoltaic detector for signal acquisition. The optical cell package contained an internal thermoelectric cooler (TEC) that controlled the temperature of the laser diode (Figure S1b). The laser TEC output (the laser temperature) was fed into a proportional-integral-derivative (PID) loop to maintain the water line absorption peak in place. The determination of $CO_2$ and COS spectra by the MIRA Pico analyser relies on referring to the position of the water spectral peak in the ~4 µm region and calculates each concentration by spectral fitting individually. Consequently, the water vapour concentration (>1000 ppm) in the air is mandatory for COS measurements, making it difficult to measure COS concentrations in low-temperature or high-altitude regions without humidification. Specifically, it is difficult to use MIRA Pico when the temperature is below -20 °C and the relative humidity is below 8% at 10 °C. The COS concentration was output from the MIRA Pico analyser in the dry mole fraction with a 1-point linear correction determined by the manufacture. Equipped with two inlets, the analyser enabled switching back and forth between the unaltered air and COS-scrubbed samples, mitigating temperature-induced drifts, and the guaranteed number of switches was 100 million. The main sources of measurement noise are due to electrical equipment, the quality of optical coatings, and purity of the laser beam. However, under the original settings, this system exhibits a significant signal drift (see section 3.1), with a standard deviation (1σ) of ± 50 ppt on a 10 min time scale, and its accuracy is also sensitive to ambient temperature. In addition, introducing high water vapour concentrations into the MIRA Pico analyser may cause condensation in the cell, making measurements impossible. Water vapour concentration was measured with the MIRA Pico. If there was not enough water vapour concentration, the water spectrum would not be fixed, and an error would occur. Therefore, you can assess whether the water vapour amount is sufficient.

We did not use a Nafion dryer to remove water; however, we used it to ensure that the water vapour concentrations of the reference and sample were the same. We added these

statements. The Nafion dryer facilitates the transfer of water molecules from a gas with a high moisture content to a gas with low moisture content via a Nafion membrane. Dry air is commonly used in counter current flows to remove water from samples. However, using a parallel flow, the water contents of the reference and sample gases were maintained at similar levels. Throughout the measurement period, water vapour concentration remained constant. Therefore, no additional measures were necessary to maintain a constant water vapour level. However, depending on water vapour variations, adjustments to the ECU temperature settings may be necessary.

To check the fraction of COS removed by the activated charcoal, we conducted an additional experiment in Sections 2.4.2 and 3.2.

From the preprint results, the reason for the poor precision may be the use of outside air rather than room air as the reference gas or the deterioration of the mirror surface condition. We have added information regarding the size of the modified MIRA Pico and whether it is portable in line 284 as follows: The MIRA Pico system can be installed in a space of 50 cm × 80 cm × 80 cm and can be separated into a refrigerator and other equipment for easy transportation.

Some explanations are needed, e.g. Why is sample and reference gases switched every 30s? Is it really necessary that frequent, is it sustainable?

**Reply:** This analyser was equipped with two inlets, allowing continuous switching between the unaltered air sample and the COS-scrubbed sample. This approach terminates temperature-induced drift. If we do not switch frequently, the COS peak will drift (See Figure S3 and S4). Therefore, frequent switching is required to measure the COS concentration using the MIRA Pico. The guaranteed number of switches was 100 million. This information has been added to line 82. Therefore, it is considered sustainable.

[Figure]

Figure S3. Standard B was injected into the MIRA Pico analyser and the MIRA Pico system for more than 1 h. (a) Temperature variation of MIRA Pico analyser, (b) output COS concentration of MIRA Pico analyser, (c) relationship between temperature and output COS concentration for MIRA Pico analyser, (d) output COS concentration of MIRA Pico system, (e) temperature variation of MIRA Pico system, and (f) relationship between temperature and output COS concentration for MIRA Pico system.

[Figure]

Figure S4. The measured value of COS using MIRA Pico. The COS concentration of the sample was obtained by subtracting the average value of the preceding and subsequent reference gases injected into port 0 from the average value of the gas injected into port 1.

Temperature stability is mentioned, but results are not shown. I suggest to add more results related to temperature stability and effects on COS concentration, at least in a supplement.

**Reply:** We have added the temperature in Figure S3.

The field measurements are not described at all in the methods section, so it is very difficult to assess their relevance.

**Reply:** We have added a description of the field measurements in the Methods Section 2.5.

**2.5  Field observations**

Continuous observations of COS and water vapour concentrations were performed at the National Institute of Advanced Industrial Science and Technology in Tsukuba, Japan (36.05° N, 140.12° E, 12 m above ground level) from 12 to 21 April 2023 (Tsukuba site in Figure S4). Tsukuba is a suburban area with low-lying land and no mountains or other obstacles to the east and south, reaching the Pacific Ocean at approximately 50 km, while the north and west are mountainous inland. Industrial areas, such as power plants, are located in southwestern. From 12 to 21 April 2023, in Tsukuba City, the sunrise time was approximately at 5:00 (Japan Standard Time (JST)), and the sunset time was approximately at 18:00 (JST). Meteorological data were obtained from the Tateno site, which is located 1 km from the Tsukuba site.

From the Allan variance plot it is clear that the low-frequency drift decreased after the modification to the measurement system. However, the high-frequency noise was increased. This is not discussed at all in the manuscript or the reasons why this increase happens. It is also unclear why the optimum integration time (for highest accuracy measurements) then changes from 10s to 40s?

**Reply:**  Thank you for your comment. The results obtained from the Allan deviation are discussed in Section 3.1.

**3.1 Allan deviation**

The Allan deviation ($\sigma^2$) plots of the MIRA Pico analyser and MIRA Pico systems are shown in Figure 2. The MIRA Pico analyser was dominated by white noise up to an integration time of 5 s and started to drift in an approximately non-linear manner after approximately 30 s. The Allan variance was the smallest at 5 s, with a value of 680 ppt$^2$. After modification, the MIRA Pico system exhibited white noise for up to 20 s and started to drift in an approximately non-linear

manner after approximately 40 s. The Allan deviation was the smallest at 40 s, at 274 $ppt^2$. The Allan deviation was lower in the MIRA Pico analyser than in the MIRA Pico system at an integration time of less than 5 s. The increase in white noise in the MIRA Pico system was due to vibrations from the refrigerator or fan or the effects of cable extensions. However, the influence of drift was smaller in the MIRA Pico system than in the MIRA Pico analyser. The decrease in the effect of drift in the MIRA Pico system was due to the stabilisation of the cell temperature, as shown in Figure S3. Although the MIRA Pico system was more stable than the MIRA Pico analyser, the MIRA Pico analyser is strongly influenced by drift (Figure S3). The COS concentrations without correction with a reference gas are shown in Figures S3c and f. Introducing references for frequent actual atmospheric measurements is necessary. In subsequent experiments, the reference gas was injected every 30 s to measure COS concentrations with high precision (low Allan deviation). To ensure data stability, the data collected 10 s after switching were disregarded. Thus, the exchange time was 10 s, reference measurement was 20 s, exchange time was 10 s, and sample measurement was 20 s, which required a total of 1 min for one cycle. The measured value was obtained by subtracting the average value of the preceding and subsequent reference gases injected into port 0 from the average value of the gas injected into port 1 (Figure S4).

Trajectory analysis would be needed to know where air parcels actually came from to better analyze and give relevance to Fig. 8. The whole field measurement section is lacking supporting information and either needs to be expanded or left out.

**Reply:** Thank you for your suggestion. We conducted a backward trajectory analysis, as shown in Figure 9, and discussed the results in Sections 2.6 and 3.5. We have also added Blake et al. (2004) as a reference.

2.6       Backward trajectory analysis

Three-day backward trajectories were analysed using the HYSPLIT (Hybrid Single-Particle Lagrangian Integrated Trajectory) model available online at www.arl.noaa.gov/ready/hysplit4.html (Rolph et al., 2017). The backward trajectories arrived at the Tsukuba site at 3:00, 9:00, 15:00, and 21:00 JST each day, and the arrival height was set at half of the planetary boundary layer. The meteorological data were obtained from a 1° × 1° grid.

[Figure]

Figure 9. Three-day backward trajectory analysis from 12 to 22 April 2023. Coloured trajectories show COS concentrations observed at Tsukuba every 3:00, 9:00, 15:00, and 21:00 (JST). (Red: ≥ 550 ppt, Yellow: 550–500 ppt, Green: 450–500 ppt, and Aqua: ≤ 450 ppt). Plots show hourly positions.

Backward trajectory analysis was performed to clarify the relationship between the air mass and COS concentration. Coloured backward trajectories according to COS concentrations are shown in Figure 9. Air masses arriving at the Tsukuba site during periods of high COS concentrations (> 550 ppt) passed southwest. There is an industrial area along the Keihin industrial zone in the southwest at the Tsukuba site. Based on observations conducted from February to April 2001, the Pacific Belt, including the Keihin industrial zone (Figure S2), was reported to be an area of high COS emissions from carbon black production, aluminium production, pigment production, sulphur recovery, and carbon disulphide ($CS_2$) emissions from rayon production (Blake et al., 2004). Once released, $CS_2$ is rapidly converted into COS and sulphur dioxide (Chin and Davis, 1993). As described above, there was an influence of diurnal variation; however, the COS

concentrations exceeding 550 ppt observed in this study were also likely increased by COS released from the Keihin industrial zone. The MIRA Pico system was used to perform atmospheric observations and provided reasonable results, indicating that this analytical method can be used to observe variations in atmospheric COS concentrations.

[Figure]

(From Blake et al., 2004 Figure 1)

Reference

Blake, N. J., Streets, D. G. Woo, J. H., Simpson, I. J., Green, J., Meinardi, S., Kita, K., Atlas, E., Fuelberg, H. E., Sachse, G., Avery, M. A., Vay, S. A., Talbot, R. W., Dibb, J. E., Bandy, A. R., Thornton, D. C., Rowland, F. S., and Blake, D. R.: Carbonyl sulfide and carbon disulfide: Large-scale distributions over the western Pacific and emissions from Asia during TRACE-P, J. Geophys. Res., 109, D15S05, doi:10.1029/2003JD004259, 2004.

Detailed comments:

Figure 1: Where is the outlet from the analyzer? Did you take the measurement cell out of the analyzer and put it in the refrigerator..? You mention in the text the size of the analyzer but what is the size of the modified setup? A picture of the setup would also be nice, e.g. in supplement

**Reply:** We have added the photos in Figure S1 and added the outlet from the analyser in Figure 1.

[Figure]

Figure 1. Schematic diagram of the modified MIRA Pico. System components: Pump, vacuum pump; cell, optical cell; ECU, electric cooler unit; T, temperature sensor; P, pressure sensor.

[Figure]

Figure S1. (a) MIRA Pico analyser, (b) optical cell package, (c) thermal insulation, and (d) MIRA Pico system in the refrigerator. The optional cell package consists of the optical cell and the upper and front base. Thermal insulation is an aerogel sheet on top of blue Styrofoam. The optical cell package is covered with insulation. The fan columns are stuck onto the blue Styrofoam.

Figure 2: Gridlines would help the reader. From low frequency variation you can determine if the drift is linear or non-linear, please do that either in the text or also show lines of linear and non-linear drift in the plot as in e.g. Gerdel et al. 2018. In the caption: "Allan deviation plots with original…"

**Reply:** We have added gridlines and showed lines of linear and non-linear drift in the plot in Figure 2, with Werle (2010) as a reference.

[Figure]

Figure 2. Comparison of Allan deviation plots of (a) the MIRA Pico analyser and (b) the MIRA Pico system as a function of the integration time, as described in Werle (2010). The red line indicates white noise, and the black dashed and solid lines indicate non-linear and linear drift, respectively.

Reference

Werle, P.: Time domain characterization of micrometeorological data based on a two sample variance, Agricultural and Forest Meteorology, Volume 150, Issue 6, 2010, Pages 832-840, https://doi.org/10.1016/j.agrformet.2009.12.007, 2010.

Figure 3: Why is there an (more or less) empty area in the middle of the scattered measurements…? As if the analyzer could not detect certain concentrations, only the scatter. Panel a COS concentrations seem to be 10 times too high. What exactly is the difference of plots a and b? Averaging? "The plot was almost every second" what does this mean? Do you mean the frequency of the measurements was 1 Hz?

**Reply:** In the COS concentration measurement results, the middle was empty because it switched between the reference and sample gases. The vertical axis represents the largest values. The raw measurement results are shown in Figure S4, and only the reference-corrected COS concentrations are shown in the new Figure 3.

[Figure]

Figure 3. Evaluation of repeatability of COS concentration using standard D or F as a sample and outside air as a reference gas. The blue lines show the overall average, and the red lines show the 15-min moving average COS concentration for standards D and F. The plot is approximately every 1 min cycle. The results for the calibration periods in which standards A to C were injected were removed.

Figure 4: Why is there data missing May 1 to May 10[th] and then again for a few days..? What is the time scale of these measurements? Please add the concentrations of the standards e.g. as lines to the plot or in the figure caption. Why are all Standards measured as 50-60 pmol mol[-1] lower than what they should be?

**Reply:** We were unable to measure the COS concentrations during the large gaps. The missing data from 1 to 9 May were due to a computer problem with the microcontroller of the MIRA Pico system. On 1 May, we applied our program to the controller to communicate with another server computer in the LAN while booting. There was an error in the program that overloaded the controller and slowed down the entire system. We observed this on 10 May, and the failed program was removed from the controller. From 11 to 16 May, measurements could not be performed because the pump was stopped.

We described these reasons in Section 2.4.3. From 1 to 9 May, COS concentrations could not be measured because of an error in our program. From 11 to 16 May, COS concentrations could not be

measured because of a MIRA Pico pump failure. This was a problem with our management and not a problem caused by MIRA Pico.

The COS concentration was lower than the standard because it showed the output of the MIRA Pico as is. The activated charcoal did not remove all the COS, but the COS concentration passing through the activated charcoal was almost constant. Therefore, the output values of all standards were lower. This is explained in Section 3.2. Using standards A to C through activated charcoal as a reference gas did not significantly change the measured COS concentrations in standard E, indicating that the reference gas remained constant even when the atmospheric COS concentration varied widely from 360 to 565 ppt. However, when the COS concentrations in standards A–C were measured using humidified $N_2$ (80%) + $O_2$ (20%) gas as a reference, the COS concentrations increased by 75–83 ppt compared to when air passing through the activated charcoal was used as a reference (Table S2). This indicates that the activated charcoal did not remove all the COS. Although the air passed through activated charcoal as a reference in this study is within the range of measurement precision, it may be recommended that air in cylinders be used as a reference for more accurate analysis.

Table S2. Removal of COS by activated charcoal.

| Standards | Difference in COS concentration (ppt) when pure air and outside air are used as reference[a] |
|:---:|:---:|
| A | 77 |
| B | 75 |
| C | 83 |

[a]The difference in COS concentration was calculated by subtracting the COS concentration when outdoor air through pump, ECU, activated charcoal, and a Nafion dryer were used as reference gases from the COS concentration when a humidified pure air through ECU and Nafion dryer were used as reference gases. The calibration curve was based on outdoor air as the reference gas.

Figure 5: Please add panels a and b, and refer to them in the caption. Why are there dots inside the circles in the lower (b) panel?   Why are error bars omitted?

**Reply:** We have added panels a and b and referred to them in the caption. We have deleted the dots in the circles in panel b. We have removed the error bar because it made it difficult to see the plot. The results of the error bar plots are presented below.

[Figure]

Figure 6: There is a big gap from May 1$^{st}$ to May 10$^{th}$ and then some days again, and after this gap there is a big step change especially for Standard A concentrations. What happens here? Please explain in the text. Why is this step change not visible in panel b? Did COS concentration also have this step change..?

**Reply:** We have described the reasons for these large gaps. We have shown the relationship between the water vapour and COS concentrations. The COS concentration did not change when the water vapour concentration changed; therefore, it is not visible in panel b. As shown in Figure 4, this gap affected the COS concentration measurement.

We have added the following to Section 3.4 for further discussion of the water vapour concentration. The water vapour concentration was higher when Standard A was measured compared to Standards B and C. This is because the standards were introduced in the order of A, B, and C, and the water in the ECU decreased as standard A, dry air, flowed through.

Figure 7: I suggest to move this fig to a supplement
Reply: We have moved it to the supplement.

Figure 8: Meteorological variables would be very beneficial in interpreting this figure. Please add at least wind speed and direction as well as air temperature and relative humidity time series plots to this figure.

**Reply:** We have added the wind speed and direction, as well as air temperature and relative humidity time series plots, to this figure.

[Figure]

Figure 7. (a) COS concentrations observed at the Tsukuba site. (b) wind speed (black) and wind direction (red), and (c) temperature (black) and relative humidity (red) observed at the Tateno site. The wind direction is shown relative to the true north.

Figure 9: Averaging time 15 min is mentioned twice, please check. It would be informative if you plot all original datapoints (maybe in lighter color) and then the averaged values on top in panel a.

**Reply:** We have removed results regarding COS concentrations observed in cars because we were unable to get any evident conclusions about this result.

Figure 10: Is "Tsukuba site" the same as "swamp"..? Please make clear and be consistent. The scale on the lower right corner should be more visible. You could mark urban areas e.g. with rectangles/circles in the maps.

**Reply:** The Tsukuba site is not a swamp. We appreciate that it is difficult to understand. We have removed this Figure.

Abstract: Mention the manufacturer (Aeris Technologies) of MIRA Pico somewhere

**Reply:** We have added it.

L25: "carbon dioxide ($CO_2$)", as this is the first time $CO_2$ is mentioned in the text

**Reply:** We have added the carbon dioxide ($CO_2$) accordingly.

L28: "…limited number of COS observation sites."

**Reply:** We rewrote this sentence.

L34: Aerodyne quantum cascade laser spectrometer (QCLS) (Aerodyne Research Inc., Billerica, USA)

**Reply:** We have revised the text accordingly.

L36: Kooijmans, not Kooijimans

**Reply:** We have revised the text accordingly.

L37: ABB-LGR off-axis integrated cavity output spectroscopy (OA-ICOS)

**Reply:** We have revised the text accordingly.

L47: "less than half that of a conventional COS analyzer": How much is it with the modifications you made?

**Reply:** We have added the price in lines 55 and 289 as follows: at the Japanese rates in 2023, an Aerodyne or LGR COS analyser would cost approximately 30 million yen, while a MIRA Pico analyser would cost approximately 7.5 million yen. At the Japanese rates in 2023, a modified MIRA Pico could be built for 9 million yen, including MIRA Pico, considering all the inlet systems except for standard gases.

L55: "carbon dioxide ($CO_2$)" -> "$CO_2$" as you should already introduce $CO_2$ in L25

**Reply:** We have revised the text accordingly.

L55: "water vapor concentrations"

**Reply:** We have revised the text accordingly. We revised this throughout the manuscript.

L59-L61: These two sentences are more like introduction than methods; suggesting to move to Introduction.

**Reply:** We have revised the text accordingly.

L62: "standard deviation (1 $\sigma$) of $\pm$ 50 pmol mol$^{-1}$ on 10 min time scale"

**Reply:** We have revised the text accordingly.

L74: What was the material of filter and inlet tubing?

**Reply:** We have added the material of the filter and inlet tubes in line 70 as follows: The upstream filter is a 0.01-micron fluorocarbon borosilicate glass microfiber element, and the tubing upstream from the optical cell is made of polyurethane tubing.

L80: Is this shown somewhere, that there is no difference? Why are pump and ECU then used if there is no change? This needs some rephrasing.

**Reply**: We have added these results to Table S1. In addition, we have added a note explaining why we used the pump and ECU in line 102, as follows: The ECU prevents water from condensing inside the cell.

Table S1. Difference of COS concentrations with or without the pump and ECU.

| Experiments | Ports | Connections | Difference (averaging and standard deviation ($1\sigma$)) |
|---|---|---|---|
| *1* | 1 | With pump +ECU | 2.1 ± 29 ppt |
| | 0 | Without pump and ECU | |
| *2* | 1 | Without pump and ECU | 1.2 ± 21 ppt |
| | 0 | With pump +ECU | |

L83: "Activated charcoal can remove a part of COS." This sounds very dangerous, why would you want to remove some of the target gas..?.

**Reply:** We apologise for the lack of clarity in this explanation. Activated charcoal was used to remove COS from the reference gases. We have rewritten the sentence in line 115 as follows: Activated charcoal showed COS removal from the reference gas.

L90: You mention when the standards were filled, but not when were the lab measurements and long-term stability tests done? From Fig 4 I see in spring 2023, but mention it also in the text in Methods section

**Reply:** We have added information about our experiments in the Methods section.

L95: If sample air and reference gas are switched every 30s and data are collected only during 10s, that means only 20s of actual data remain every minute..?

**Reply:** Thank you for the comment. We have added the following sentence in line 199: the exchange time was 10 s, the reference measurement was 20 s, the exchange time was 10 s, and the sample measurement was 20 s, for a total of 1 min.

L104: ".. many studies have reported that.." please provide references

**Reply:** We have added the references accordingly.

L115-116: "some level of fluctuation", please quantify how much (e.g. 1 min std). What time frame is the std for standard C presented?

**Reply:** We conducted an additional experiment and added specific information (please check Figure 3 and Section 3.2).

**Reply:** We have revised the text accordingly.

L135-139: It was mentioned previously that a water vapor concentration of at least 5000 µmol mol$^{-1}$ is needed for COS measurements, but you report a water content of 4000 µmol mol$^{-1}$ for Tsukuba. Are the COS measurements then unreliable? How is the water vapor concentration measured after humidifying and drying the air?

**Reply:** Even if the water concentration was below 5000 ppm, the COS concentration could be measured because the water was humidified by the ECU. As shown in Figure 6, when the NOAA standard gas was injected, the water vapour concentration exceeded 5000 ppm. Figure S5 shows the water vapour concentration during the atmospheric observation period, which always exceeded 5000 ppm. Considering these facts, all data presented here are reliable for water vapour concentrations above 5000 ppm.

[Figure]

Figure S5. Water vapour concentrations from the observation period of the COS concentration at the Tsukuba site.

L148: "Koiijimans" -> "Kooijmans"

**Reply:** We have revised the text.

L160-164: The amount of reference/calibration gas used depends on the user and target of the measurement (e.g., frequent calibrations are not as necessary for flux measurements as they are for accurate atmospheric concentration measurements), not only on the analyzer used. Kooijmans et al. (2016) measured a reference gas every 30min for 3min, not for 10min, so this estimate of their reference gas use is quite misleading.

**Reply:** We apologise for this misleading sentence. We have deleted the comparison of gas consumption because it depends on the study.

Sect. 4: I suggest to rethink the organization of the sections, since this section is still very much about results and discussion (sect. 3). I suggest to change the numbering of this section from 4 to 3.5 and the subsections as 3.5.1 and 3.5.2.

**Reply:** We have changed the sections.

L186-188: Sentence beginning with "They decreased.." and the following sentence: Please rephrase these sentences as they are not very clear. One suggestion would be "COS concentrations increased after sunrise until approximately 16:00, after which they decreased." I would also suggest a plot with average/median diurnal variation. It is quite difficult to determine from Fig. 8; e.g., it seems on 19th April the decrease would happen at 18:00 while on 20th April it happens only after midnight.

**Reply:** We have rephrased the sentence as follows: The average and median COS concentrations decreased from night to dawn and then increased from dawn to 16:00 (Figure 8). Additionally, we have added a plot showing the average/median diurnal variation in Figure 8.

[Figure]

Figure 8. Diurnal variation of COS concentration at the Tsukuba site. The plots represent the difference from the first plot of the day (circle: average, cross: median).

L192-194: The decrease of the atmospheric concentration is especially because of the atmospheric mixing conditions, and since you observe a decrease during nighttime it means there is a nighttime sink in the ecosystem (e.g. soil bacteria as you suggest). Concentrations increase again after sunrise as the atmospheric boundary layer increases and mixing layer develops, mixing higher concentrations from above with the air close to surface.

**Reply:** We have rewritten the sentences in line 307 as follows: The decrease in COS at night can be thought to be mainly caused by ecosystems such as soil bacteria, as reported by Kato et al. (2008) and Kamezaki et al. (2016). The increase in COS concentrations after sunrise is suggested to be due to atmospheric boundary layer increases and mixing layer develops, mixing higher concentrations from above with the air close to surface (Campbell et al., 2017).

L199: I would also suggest flux measurements (either by chambers or eddy covariance) to determine the sinks and/or sources.

**Reply:** We have added the flux measurements.

L207: "Almost the same COS concentration was observed": please elaborate this, especially close to swamp the concentration is considerably different

**Reply:** We have deleted the sentence.

 Where are the urban areas located? Not really visible from the map.

**Reply:** We apologise for the difficulty in understanding the information presented in the figure. We have deleted this sentence accordingly.

L214: Since the wind was from south-southwest, could it be there is COS signal from the industrial area?

**Reply:** Considering the wind direction, we can interpret it in that manner, but there were fewer observations, and the interpretation was not exact. Therefore, we deleted this sentence.

L214: Why 14:00-14:20 is selected as an interesting timeframe?

**Reply:** We have deleted this sentence.

L225: Photoproduction from wetlands but also consumption by photosynthesis has been reported in previous studies (see synthesis study by Whelan et al., 2018)

**Reply:** Thank you for your comment. As you have pointed out, photoproduction from wetlands can produce COS in swamps. However, it was difficult to identify the causes of COS concentration fluctuations during this study.

L235: Why only areas with high or low COS concentration are interesting? How to even know that without measuring?

**Reply:** We have deleted this sentence. The purpose of creating these sentences was as follows: When applied to areas other than those with high or low COS concentrations, fixed-point observations may be more useful than onboard equipment. As we received a comment from another reviewer, the disadvantage of vehicle observations is that it is difficult to evaluate the relationship with the weather field compared to fixed-point observations. To draw evident conclusions from areas where analysis is difficult, we believe that it is necessary to move to areas where changes in concentration are large. To predict whether the COS concentration will be high or low without taking measurements, it is necessary to make assumptions based on information, such as whether the factory uses sulfur or the results of satellite observations and models.

L239: Allen -> Allan

**Reply:** We have revised the text.

L240: Allan variance was decreased only at low frequency, but at high frequency it actually increased!

**Reply:** We have rephrased it.

**Reply:** We have added the information in line 69 as follows: The MIRA Pico analyser is ultracompact (30 × 20 × 10 cm) and lightweight (3 kg), uses a small cell (60 mL), and has an optical path length of 13 m.

**Reply:** We have deleted this sentence.

**Reply:** We have described it above.

**Reply:** We have revised the sentence.

---

## Author Comment (AC2)

**Blue: Referee comment**

Black: Our comment Red: The sentences in our manuscript

**Overall evaluation and general comments**:**

The preprint by Kazuki Kamezaki on a "compact continuous measurement system for atmospheric carbonyl sulfide" describes and evaluates modifications to improve a commercially available OCS instrument. In that respect, the term "Development of..." in the title is clearly an overstatement. While this could be amended by changing the title, I 'm afraid that the paper in its present form contains too little substance and too many errors and ambiguities to be publishable in AMT. The necessary modifications go beyond normal revisions, which is why I recommend not to accept this paper for publication. Below, I explain my main concerns. Then I list some specific and technical issues to underline my criticism and to provide recommendations should the authors chose to rewrite the article for resubmission at a later stage.

**Reply:** Thank you for your comment. We have removed the "development of" from the title. We apologise for the poor quality of our preprints. We have considered your comments seriously and revised the manuscript accordingly. Thank you for your comments.

The methods section (Section 2) falls short of providing anything close to a comprehensive instrument description. Neither is the overall measurement and data analysis concept of the original MIRO analyzer fully described, nor are shortcomings or problems that motivated the adaptations made to the analyzer in this work properly characterized. In a methodological study such as this one, I expect, for example, the following details to be given and explained: What are the dimensions of the optical cell and what is the absorption path length? What light source and detector are being used? At what wavelength is OCS being measured? Does the analyzer measure an absorption spectrum that is analyzed by spectral fitting, or does it measure absorption only at one or more distinct wavelengths? Which water absorption band is used as a reference peak, and how does this referencing work? How is the temperature of the light source stabilized, and how is wavelength and baseline stability ensured? What are the main sources of noise? Without such methodological details, it is impossible to put the applied modifications into context and evaluate their benefits as well as the overall performance measures.

**Reply:** Thank you for your comment. The manufacturers were contacted for further information and provided as much as they could about the MIRA Pico without revealing some of their trade secrets, according to them. We have added the following information to Section 2.1: COS concentrations were measured using a MIRA Pico analyser (Figure S1a). The MIRA Pico analyser is a portable, field-deployable instrument designed for the simultaneous measurement of

COS, CO2, and water vapour concentrations with one Hz. The MIRA Pico analyser is ultracompact (30 cm  $\times$  20 cm  $\times$  10 cm), and lightweight (3 kg), uses a small cell (60 mL), and has an optical path length of 13 m. The upstream filter is a 0.01-micron fluorocarbon borosilicate glass microfiber element, and the tubing upstream from the optical cell is made of polyurethane tubing. The flow rate was adjusted using the pump voltage, which was set at approximately 210 mL min-1. The pressure inside the optical cell was 140 mbar. The analyser incorporates a midinfrared tuneable diode laser and a HgCdTe (MCT) photovoltaic detector for signal acquisition. The optical cell package contained an internal thermoelectric cooler (TEC) that controlled the temperature of the laser diode (Figure S1b). The laser TEC output (the laser temperature) was fed into a proportional-integral-derivative (PID) loop to maintain the water line absorption peak in place. The determination of CO2 and COS spectra by the MIRA Pico analyser relies on referring to the position of the water spectral peak in the  $\sim 4 \mu m$  region and calculates each concentration by spectral fitting individually. Consequently, the water vapour concentration (>1000 ppm) in the air is mandatory for COS measurements, making it difficult to measure COS concentrations in low-temperature or high-altitude regions without humidification. Specifically, it is difficult to use MIRA Pico when the temperature is below -20 °C and the relative humidity is below 8% at 10 °C. The COS concentration was output from the MIRA Pico analyser in the dry mole fraction with a 1-point linear correction determined by the manufacture. Equipped with two inlets, the analyser enabled switching back and forth between the unaltered air and COS-scrubbed samples, mitigating temperature-induced drifts, and the guaranteed number of switches was 100 million. The main sources of measurement noise are due to electrical equipment, the quality of optical coatings, and purity of the laser beam. However, under the original settings, this system exhibits a significant signal drift (see section 3.1), with a standard deviation  $(1\sigma)$  of  $\pm$  50 ppt on a 10 min time scale, and its accuracy is also sensitive to ambient temperature. In addition, introducing high water vapour concentrations into the MIRA Pico analyser may cause condensation in the cell, making measurements impossible.

With respect to performance measures (given in Section 3) some of the numbers are neither well rationalized nor convincing. The terms accuracy, precision, and uncertainty are not always used in a correct and consistent manner, and it is not at all clear how the results of the various measurement series performed on know standards are used to validate or correct OCS concentrations measured in the field.

**Reply:** We apologise for not providing sufficient explanation. We have added the experimental methods to the Methods section of Section 2.4.2. We have also added the results of our tests on the stability of the reference gas in Section 3.2, as we thought you were probably concerned that the reference gas was not stable. Sections 2 and 3 describe each analysis and its use.

The description of the field measurements in Section 4 is not very meaningful in my opinion. The explanations and interpretations are largely rudimentary and speculative. They go too far for a purely technical paper, but not far enough in order to be of scientific value. Given the time period reported and the region covered, the results are only of (rather limited) local interest, and the case that more such measurements could be a game- changer in understanding the OCS budget and cycling is not convincing.

**Reply:** The conclusion has been overstated. We have removed the results regarding the COS concentrations observed in cars. However, we have discussed the results of 10 days of observation at the Tsukuba site based on meteorological data. Because we wanted to show that reasonable COS concentrations could be observed with the MIRA Pico system in this field observation, we changed our conclusions as follows: The MIRA Pico system was used to perform atmospheric observations and provide reasonable results, indicating that this analytical method can be used to observe variations in atmospheric COS concentrations.

**Specific comments**:**

Line 27 – 30: In my opinion, these two sentences are misleading. First, there are no "local contributions to SSA production", which happens in the stratosphere and is not directly connected to the near-surface OCS cycling. What you mean is contributions to the budget, which in turn plays a role for how much OCS reaches the stratosphere, where SSA is produced. More importantly, the statement that "tropospheric OCS sources and sinks entail great uncertainty due to the limited number COS observation sites" is far too simplistic. Arguably, a few more sites with OCS observations in well chosen locations could help to better quantify the regional distribution of certain sources and sinks. But overall, a great deal of understanding tropospheric OCS cycling has been achieved with data from available networks and satellites, and I expect a bit more in terms of strategy to address remaining uncertainties than simply calling for more OCS observations.

**Reply:** Thank you for your comment. The concentration of COS in the stratosphere is not uniform, as reported by Glatthor et al. (2017). The reason for this nonuniformity is thought to be the contribution of local COS emissions. Therefore, it is important to understand the distribution of local COS concentrations. As you have pointed out, increasing the number of observation points is too simple. To accurately understand the dynamics of COS, it is essential to observe ground-level concentrations of COS. However, long-term ground-based observations of COS concentrations are limited to the Americas, Europe, and Antarctica, and the long-term observation sites are biased. Models and satellite observation results cannot capture short-term fluctuations, and ground observations are considered essential for clarifying the causes of COS fluctuations

occurring at the site. Thus, we have changed the sentences in lines 29-34 as follows: COS concentrations have been monitored over the long term by National Oceanic and Atmospheric Administration (NOAA) (Montzka et al., 2007). The dynamics of the global COS are beginning to be understood through satellite observations and models (Kuai et al., 2015; Glatthor et al., 2017; Ma et al., 2023; von Hobe et al., 2023). However, the observation sites are highly biased, and there is no continuous observation data exist for Asia, South America, Africa, and the Atlantic Ocean. Observing COS at locations with no such measurement points and analysing COS variation factors results in a more accurate understanding of the COS budget. We added new references to the revised manuscript.

(From Figure 12 in Glatthor et al. (2017))

**References**

Glatthor, N., Höpfner, M., Leyser, A., Stiller, G. P., von Clarmann, T., Grabowski, U., Kellmann, S., Linden, A., Sinnhuber, B.-M., Krysztofiak, G., and Walker, K. A.: Global carbonyl sulfide (OCS) measured by MIPAS/Envisat during 2002–2012, Atmos. Chem. Phys., 17, 2631–2652, https://doi.org/10.5194/acp-17-2631-2017, 2017.

Kuai, L., Worden, J. R., Campbell, J. E., Kulawik, S. S., Li, K.-F., Lee, M., Weidner, R. J., Montzka, S. A., Moore, F. L., Berry, J. A., Baker, I., Scott Denning, A., Bian, H., Bowman, K. W. Liu, J., and Yung, Y. L. :Estimate of carbonyl sulfide tropical oceanic surface fluxes using Aura Tropospheric Emission Spectrometer observations, J. Geophys. Res. Atmos., 120, 11,012–11,023, doi:10.1002/2015JD023493, 2015.

Ma, J., Remaud, M., Peylin, P., Patra, P., Niwa, Y., Rodenbeck, C., Cartwright, M., Harrison, J. J., Chipperfield, M. P., Pope, R. J., Wilson, C., Belviso, S., Montzka, S. A., Vimont, I., Moore, F.,

Atlas, E. L., Schwartz, E., and Krol M. C.: Intercomparison of atmospheric carbonyl sulfide (TransCom-COS): 2. Evaluation of optimized fluxes using ground-based and aircraft observations. Journal of Geophysical Research: Atmospheres, 128, e2023JD039198. https://doi.org/10.1029/2023JD039198, 2023.

von Hobe, M., Taraborrelli, D., Alber, S., Bohn, B., Dorn, H.-P., Fuchs, H., Li, Y., Qiu, C., Rohrer, F., Sommariva, R., Stroh, F., Tan, Z., Wedel, S., and Novelli, A.: Measurement report: Carbonyl sulfide production during dimethyl sulfide oxidation in the atmospheric simulation chamber SAPHIR, Atmos. Chem. Phys., 23, 10609–10623, https://doi.org/10.5194/acp-23-10609-2023, 2023.

Lines 39 + 47: The statement "*However, these devices are large, expensive, and costly to maintain.*" is too unspecific as these adjectives are somewhat relative. Also, a few lines below, you define a "*good precision*" out of the blue. I would like to see more compelling arguments what the precision needs to be to tackle relevant science questions, and what the exact problems are with the size and costs of existing analyzers. In other words, where are the real problems that need to be solved? Later, it is stated that the MIRO is "*less than half the price*" of other analyzers and "*small*". Again, please be more specific here (at least make some reference to the details given in Section 2) and clearly state why the differences matter.

**Reply:** We apologise for the lack of specific information. We have rewritten the sentence in line 44 as follows: However, the problem is that these devices are large and consume much electricity. The original Aerodyne COS analyser uses 250 W (without a pump), and the original LGR COS analyser uses 400 W. These must be installed at locations where commercial power is available. However, when observing remote locations (such as forest sites in Southeast Asia and remote islands), the power supply may not always be sufficient.

In addition, we have added the following in line 56: Furthermore, the MIRA Pico analyser consumes only 15 W, making it suitable for use in remote areas where the power supply is insufficient.

Line 58 - 59: If > 5000 ppm of water vapour are needed for the OCS measurement to work, then the instrument will be useless in cold or high-altitude environments (unless you go through the not simple efforts to moisten the sampled air). This should be clearly stated.

**Reply:** We have rewritten the sentence in line 76 as follows: Consequently, the water vapour concentration (>1000 ppm) in the air is mandatory for COS measurements, making it difficult to measure COS concentrations in low-temperature or high-altitude regions without humidification. Specifically, it is difficult to use MIRA Pico when the temperature is below -20 °C, and the relative humidity is below 8% at 10 °C.

By asking the manufacturer, they answered that they could confirm a water peak up to approximately 1000 ppm. Thus, we changed the value from 5000 to 1000 ppm.

Line 66 – 69: From the text and Figure 1, I find it difficult to understand how the temperature stabilization was really done. Can you give details on the "*commercial refrigerator*"? And was the entire MIRA instrument put into the refrigerator or just the optical cell? I also don't understand why you would set the refrigerator to 15 °C and the Peltier cooler to 29 °C? What is the target temperature inside the optical cell, and is temperature monitored inside the cell or just inside the refrigerator or at the Peltier cooler? And what is the "*cushioning material*" and what is its purpose? If it is thermal insulation, then "*cushioning*" is the wrong word.

**Reply:** We have added photos to clarify temperature stabilization and the actual temperature changes to Figure S1. We have also changed the "cushioning" to an aerogel sheet. We have added information regarding the refrigerator and cushioning material in line 93. Inside the refrigerator, the optical cell package was surrounded by thermal insulation, which is an aerogel sheet on top of blue Styrofoam (Figure S1c), and the temperature was controlled using a Peltier cooler set at 29 °C placed at the bottom (Figure 1). The refrigerator releases heat from inside to outside, and the Peltier cooler under the cell regulates this heat. Double insulation with refrigerators and insulation minimises temperature fluctuations, and an aerogel sheet is interposed between the Peltier cooler and optical cell to further reduce temperature variations.